# Association between Weather Types based on the Spatial Synoptic Classification and All-Cause Mortality in Sweden, 1991–2014

**DOI:** 10.3390/ijerph16101696

**Published:** 2019-05-14

**Authors:** Osvaldo Fonseca-Rodríguez, Erling Häggström Lundevaller, Scott C. Sheridan, Barbara Schumann

**Affiliations:** 1Department of Epidemiology and Global Health, Umeå University, 901 87 Umeå, Sweden; barbara.schumann@umu.se; 2Centre for Demographic and Ageing Research, Umeå University, 901 87 Umeå, Sweden; erling.lundevaller@umu.se; 3Department of Geography, Kent State University, Kent, OH 4242, USA; ssherid1@kent.edu

**Keywords:** all-cause mortality, Spatial Synoptic Classification, oppressive weather types, distributed lag non-linear models, Sweden

## Abstract

Much is known about the adverse health impact of high and low temperatures. The Spatial Synoptic Classification is a useful tool for assessing weather effects on health because it considers the combined effect of meteorological factors rather than temperature only. The aim of this study was to assess the association between oppressive weather types and daily total mortality in Sweden. Time-series Poisson regression with distributed lags was used to assess the relationship between oppressive weather (Dry Polar, Dry Tropical, Moist Polar, and Moist Tropical) and daily deaths over 14 days in the extended summer (May to September), and 28 days during the extended winter (November to March), from 1991 to 2014. Days not classified as oppressive weather served as the reference category. We computed relative risks with 95% confidence intervals, adjusting for trends and seasonality. Results of the southern (Skåne and Stockholm) and northern (Jämtland and Västerbotten) locations were pooled using meta-analysis for regional-level estimates. Analyses were performed using the *dlnm* and *mvmeta* packages in R. During summer, in the South, the Moist Tropical and Dry Tropical weather types increased the mortality at lag 0 through lag 3 and lag 6, respectively. Moist Polar weather was associated with mortality at longer lags. In the North, Dry Tropical weather increased the mortality at shorter lags. During winter, in the South, Dry Polar and Moist Polar weather increased mortality from lag 6 to lag 10 and from lag 19 to lag 26, respectively. No effect of oppressive weather was found in the North. The effect of oppressive weather types in Sweden varies across seasons and regions. In the North, a small study sample reduces precision of estimates, while in the South, the effect of oppressive weather types is more evident in both seasons.

## 1. Introduction

There are numerous studies that have evaluated the negative impact of extreme ambient temperature on human health [1,2,3,4,5,6]. Research has used temperature itself as a predictor variable, as well as one of a number of apparent temperature metrics that also accounts for other ambient meteorological factors, such as humidity and wind speed [7,8]. In scientific literature, mortality is the most common evaluated health outcome. It has been shown that heat exacerbates numerous chronic health problems; typically either all-cause mortality or a range of broad causes such as cardiovascular or respiratory disease are studied [7]. 

In general, heat produces an immediate effect, causing health problems, such as heat syncope, cramps, exhaustion, heat stroke, and death [9,10]. Low temperature, on the other hand, is associated with a more delayed effect, showing an increase of deaths several days or weeks after exposure [6,11]. Cold-related deaths are generally due to respiratory and cardiovascular diseases [5].

The combination of high ambient temperature and high humidity produces an increase in mortality and morbidity because of the stress of the human thermoregulatory system [12]. At high temperatures, high relative humidity reduces evaporative heat loss, while evaporative heat loss is increased when relative humidity is low [13]. On the other hand, low ambient temperature and low relative humidity favour the transmission of the influenza virus via aerosols [14]. Carder et al. [15] showed mixed results about the effect of wind speed on mortality. In addition, Ferrari et al. [16] showed that solar radiation, air pressure, and wind speed have an influence on respiratory health similar to that of temperature.

Humans respond to the collective effect of all ambient weather conditions rather than to isolated factors [17]. Thermal perception is comprised of both thermal sensation, changes in the environment (e.g., temperature) detected on the skin surface [18], and thermal comfort, the psychological state accounting for the level of satisfaction with the surrounding environmental conditions [19]. To estimate the impact of weather on human health, common metrics incorporate the apparent temperature, which includes wind speed and humidity in addition to temperature, to consider the effect on thermoregulation [20]. Therefore, one effective approach is to model the impact of weather on mortality holistically. Based on this approach, daily weather can be categorized into “weather types”, combining temperature with other meteorological factors, such as wind speed and humidity [17,21].

One such weather type classification is the Spatial Synoptic Classification (SSC), which classifies every day at a given location into one of seven weather types: Dry Polar (DP), Dry Moderate (DM), Dry Tropical (DT), Moist Polar (MP), Moist Moderate (MM), Moist Tropical (MT), and Transition (TR) [22]. The SSC method was initially developed by Kalkstein et al. (1996) [23] and later revised and refined by Sheridan (2002) [22]. These classifications are based on several weather variables: Temperature, dew-point temperature, pressure, wind speed, and cloud opacity. A more detailed explanation of the methodology can be found in Sheridan (2002) [22].

The SSC is a relative (rather than absolute) classification system [21,22]. The relationship between weather and health varies geographically [24,25] and in time across seasons [21,22], taking into consideration human thermal acclimatization and adaptation [26]. This is an important advantage of using the SSC. When the relationship between weather and health is assessed, the most common approach is to investigate the health effects of a single variable (e.g., air temperature) [21,26], but the SSC approach presupposes that weather affects human health as a whole. Hence, to consider all environmental factors and their interactions, it is crucial to assess the relationship between human health and the weather, taking into consideration the mechanisms related to physiological response [26].

The SSC has been used in a wide range of research studies, mainly in health related studies but also in assessing climate change [21], and it has been utilized in numerous heat-health warning systems in the USA and several countries around the world [27,28]. This is the first study assessing the association between weather types of SSC and all-cause mortality in Sweden, and it could be useful in improving the national health early warning system.

In Sweden, increased mortality is associated with lower temperatures in winter and with higher temperatures in summer [29,30,31]. However, it is important to study not only the impact of temperature but also that of the different air masses or weather types on mortality. Thus, the aim of this study was to assess the association between specific weather types that negatively affect human health (oppressive weather types) and daily total mortality in two northern and two southern locations in Sweden. These four locations have different climatic conditions, giving a good idea about regional variations.

## 2. Materials and Methods 

The study period was set from 1 January 1991 to 31 December 2014. Daily mortality and yearly population data from south-west Skåne (Scania) county (21 municipalities), Stockholm county (26 municipalities), Jämtland county (12 municipalities), and eastern Västerbotten county (5 municipalities) (Figure 1, Appendix A) were obtained from the Linnaeus database at the Centre for Demographic and Ageing Research (CEDAR), Umeå University, Sweden. The information of daily deaths is based on the place of residence. This database provides national, health-related, demographic, and lifestyle variables for the Swedish population [32]. The daily number of inhabitants per location during the studied period was estimated using linear interpolation.

Hourly and daily measurements for temperature, wind speed, precipitation (rain- and snowfall), cloud cover, dew point, and air pressure were obtained from the Swedish Meteorological and Hydrological Institute (SMHI). Specifically, meteorological data were obtained from the following weather stations: Malmö A (Skåne), Bromma airport (Stockholm), Östersund (Jämtland), and Umeå airport (Västerbotten). Where data were missing, to improve completeness of data, meteorological observations were taken from adjacent stations located no more than 100 km from the main weather station but inside the correspondent study location, and for missing hourly data, interpolation was done from adjacent hours.

The meteorological data were used to categorize the daily weather for each location into one of the seven weather types according to the Spatial Synoptic Classification (SSC) [22]. Meteorological parameters (temperature, dew point, wind speed and direction, air pressure at sea level, and cloud cover) measured every six hours were used to categorize the daily weather. The classification distinguishes between the following weather types: Dry Polar (DP), Dry Moderate (DM), Dry Tropical (DT), Moist Polar (MP), Moist Moderate (MM), Moist Tropical (MT), and Transition (TR) [22]. It identifies weather types on spatio-temporally relative scales; that is, the character of each weather type will vary by location and time of year. Table 1 shows the mean frequency of days (%), air temperature at 03h (Ta), air temperature at 15h (Tp), and dew point at 15h (Td) associated with each SSC type for January and July at each of the four locations.

### Statstical Analysis

To assess the effect of the weather types on mortality, seasonal time series datasets were created for each location as follows: Extended summer (May–September; hereafter, “summer”) and extended winter (November–March, hereafter, “winter”). Subsets were analyzed using a time series quasi-Poisson regression using the distributed lag non-linear models [33] package *dlnm* in R. A quasi-Poisson model was used to account for over-dispersion by fitting an extra dispersion parameter. A distributed lag non-linear model (DLNM) accounts for the nonlinearity of the exposure–response dependencies and the delayed effects of weather types on mortality by determining the change in mortality during a specified period after the exposure to the oppressive weathers [34]. DLNM uses a cross-basis function to represent the delayed non-linear exposure–outcome relationships [33].

In our analysis, we assessed the lag structure of the effects and the cumulative impact of weather types on mortality over 14 days in summer and 28 days in winter, with knot placements for the lags at three equally spaced positions. The *logknots* function (from the *dlnm* package) defines knots for lag space at equally-spaced log-values, and it was expressly created for lag–response functions [35].

In order to simplify the interpretation and to facilitate the comparison among weather types, seasons, and locations, only one standard model was used. Different combinations of equally spaced knots in the cross-basis matrix and degrees of freedom for seasonal and long-term trends were tested. The quasi Akaike information criterion (qAIC) of every combination of knots and degrees of freedom (df) for each season was summed over the four locations (see Appendix A and Appendix A). The model with the lowest summed qAIC for both seasons had three knots and three df for the seasonal and long-term trend.

The model used in our study was as follows:

Log(outcome) = intercept + DP + DT + MP + MT + ns(year) + ns(DOY) + DOW + outliers.

The model outcome is all-cause daily deaths. To assess the association between the oppressive weather types and daily mortality, binary variables were created for each of the four types considered (Dry Polar—DP, Dry Tropical—DT, Moist Polar—MP, and Moist Tropical—MT), and days not considered as “oppressive” (Dry Moderate, Moist Moderate, and Transition) served as the reference category. A natural spline (ns) was used to control for long-term trends (year) with three degrees of freedom. To describe the seasonal effect within each year, a natural spline was fit to the days of the year (DOY), with three degrees of freedom. In addition, a categorical variable DOW (day of the week) was included. Outliers in Stockholm (excess of deaths) were incorporated into the model as a categorical variable using “1” for dates with extreme death counts of the Swedish population due to large disasters (Sinking of the “Estonia” ferry, September 1994, and the Tsunami in Indonesia, December 2004) and “0” otherwise. The interpolated daily population under risk was used as the offset variable. We computed relative risks (RR) of mortality with 95% confidence intervals (CI) for each weather type. 

In the next step, relative risks were pooled to a northern region (Västerbotten and Jämtland) and a southern region (Skåne and Stockholm), respectively, using multivariate meta-analysis with the *mvmeta* package [36]. All statistical analyses were performed using R statistical software [37].

## 3. Results

Summary statistics of every study site are shown in Table 2. Moist Moderate (MM) was the most frequent weather type across the four sites, showing the highest frequency of the period in Skåne (38.9%) compared to the other three places. The least frequent weather type in all locations was Dry Tropical (DT), with the lowest frequency in Västerbotten (2.2%).

The definition of these weather types is relative to the season and the study location. During summer in the southern locations, the DT weather type (Mean July temperature and dew point at 1500 26 °C and 11 °C) was associated with a significant increase of all-cause mortality, with an RR of 1.02 (1.00–1.04) at lag 0, and with significantly increased risk until lag 6; MT (24 °C and 15 °C) showed an RR of 1.03 (1.00–1.05) at lag 0 and was significant throughout three days. Moreover, in southern locations, MP (15 °C and 11 °C) weather significantly increased mortality from lag 4 to lag 11; however, the RR increase was very small, ranging from an RR of 1.004 to one of 1.006. On the other hand, in the northern study locations, only the DT (25 °C and 10 °C) type was associated with mortality at lag 0 (RR = 1.03, 1.00–1.06), but this was counteracted by a significant reduction of mortality at lag 10 and 11. It is interesting to highlight the effect of DP (14 °C and 5 °C) during the summer in northern locations where mortality decreased considerably at short lags (lag 0; RR = 0.97, 0.93–1.01), although the impact was not statistically significant (Figure 2).

In the winter, no significant effect was found in the northern locations; The RR of the tropical types could not be estimated because of their rare occurrence at that time of the year. In the southern locations, the cold DP (Mean January temperature and dew point at 1500 −7 °C and −10 °C) and MP (−4 °C and −5 °C) produced a small albeit significant increase in mortality from lag 6–10 with an RR from 1.002 to 1.004, and from lag 19 to lag 26 with an RR between 1.001 and 1.003, respectively. Likewise, DT increased the relative risk at longer lags (lag 19–lag 20) (Figure 2). More characteristics of each weather type are shown in Table 1.

The cumulative effect was estimated as the sum of all lagged effects of each oppressive weather type through lag 14 in summer and to lag 28 in winter. The cumulative impact of the oppressive weather types on all-cause mortality was stronger in southern locations, particularly in summer when the cumulative relative risk of DT and MT was 1.08 (1.02–1.14) and 1.05 (1.01–1.10), respectively, over the 14-day period (Table 3). However, the moist and cold MP also increased relative risk (1.05, 1.01–1.09).

In winter, none of the weather types significantly increased or decreased the cumulative risk of mortality in the northern locations in either season. However, in the south, DP was the only oppressive weather type that significantly increased the cumulative relative risk (1.05, 1.01–1.09) over 28 days. Interestingly, the DT weather showed a very high increase of mortality with an RR of 1.40 (0.87–2.23), but this was not statistically significant. (Table 3).

The results in the northern locations could be affected by the small sample size because these are much less populated areas than the southern locations.

## 4. Discussion

To our knowledge, this is the first time the SSC has been used in Sweden to estimate the impact of weather types on daily total mortality. The SSC is a useful tool for assessing the effects of weather on health because it provides an estimate of the combined effect of all variables that comprise each weather type rather than isolated factors. The physiological response is related to the whole set of weather conditions simultaneously, rather than to specific meteorological variables (e.g., temperature) [38]. The SSC can be used to estimate the local and seasonal response to stressful weather types because the weather is classified at a spatiotemporally relative scale, and it allows taking into account thermal acclimatization and adaptation [22,26].

### 4.1. Summer

In our study, the hot weather types (DT and MT) in summer were associated with an immediate increase in all-cause mortality in southern locations (Skåne and Stockholm) that remained significant for several days, and the cumulative effect over 14 days was significant for both hot weather types. Previous studies have demonstrated the association between DT and MT weather and an increasing mortality rate throughout different geographic areas [39,40,41,42]. According to Sheridan and Kalkstein [28], DT and MT generally lead to the greatest increase in the risk of death during summer. Similar results were obtained by Lee et al. [40], showing the negative impact of DT and MT+ and MT++ on mortality. MT+ and MT++ are subdivisions of the MT weather type that are hotter and more humid [22,28]; these subsets were not included in the present study since MT is less common in Sweden and so subdivisions were not feasible. 

Interestingly, a previous multicountry study carried out by Gasparrini et al. [43] showed only a small, non-significant association between mortality and high and low ambient temperature for Stockholm. In our study, however, DT and MT were associated with increased mortality in the southern locations Stockholm and Malmö. This indicates the importance of considering not only temperature, but other meteorological variables used in the classification of daily weather by the SSC methodology.

In our northern locations of Västerbotten and Jämtland, in summer, only DT weather caused a significant rise in mortality at lag 0, a shorter effect compared to the southern locations. A significant reduction of mortality occurred at longer lags, indicating mortality displacement. This effect refers to a decrease in the expected number of deaths after a harmful (e.g., heat) event when a significant number of critically ill people die that would have died shortly afterwards anyway [44]. Mortality displacement has been reported in several previous studies from several geographical areas such as Brazil [45], China [46], the United States [47], and several European countries [48]. In our northern locations, despite the high mortality at lag 0 associated with DT in summer, the cumulative effect at day 14 was not significant, possibly due to mortality displacement at lags 10–11.

Heat-related deaths are more common among vulnerable populations such as the elderly [7,49] or young children [7], as well as those with a chronic cardiovascular or respiratory illness [39,50,51]. Air pollution (PM_2.5_, PM_10_, O_3_, etc.) produces a negative health impact, including mortality [52,53,54], and is a potential confounder, being associated with days with DT and MT weather types, particularly DT, with its high ambient temperatures and extensive solar radiation [55]. Dry and hot weather (DT) can favor hyperthermia and dehydration due to increased evaporation [56]. Moreover, hot and humid weather (MT) affects the physiological mechanism of the organism to regulate temperature through perspiration and vasodilatation [20].

Interestingly, we found an association of MP with mortality during summer only in the south, where the summer is hotter than in the north. The arrival of cold air masses, an occurrence that is not rare in this latitude even in summer, could increase mortality, because according to De Freitas and Grigorieva [57], acclimatization from warm-to-cold produces a higher thermo-physiological strain than the transition from cold-to-warm. 

### 4.2. Winter

Our results showed a significant increase of mortality at longer lags during winter associated with DT at the southern sites. Conversely, Kalkstein et al. [41] found a significant effect of DT increasing mortality during the winter in Los Angeles, U.S., but this effect was at lag 0, demonstrating the impact of unusual hot weather in the cold season. Other previous research also showed that DT contributed to excess winter mortality in areas of the Southwest and on the West coast of the United States because in this region the temperatures can get very warm in winter [58]. The high risk associated with DT at longer lags in the southern sites during winter season in Sweden could be due to the reduction of mortality at shorter lags, indicating a reverse harvesting effect; however, we must consider the low precision of the estimation showing wide confidence intervals. The cumulative RR over 28 days of DT was the highest compared to other oppressive weather types but was not statistically significant (1.40, 0.87–2.23) due to the very low sample size of such wintertime events.

During winter, DP increased the mortality at longer lags in southern study locations, but not in the northern ones. This finding is consistent with previous results from Europe, reporting that cold weather affects the population more in southern than in northern locations [48,59]; similarly, in the United States, research has shown that cold days produce a greater effect in warmer than in colder climates [60,61]. Low temperatures increase mortality risk mainly due to cardiovascular and respiratory problems [7,48]. Kalkstein and Greene [58] showed that while DP and MP increase mortality in certain regions in winter, the effect is much smaller compared to the effect of DT and MT in summer.

It has been demonstrated that cold weather conditions increase mortality mainly due to cardiovascular, respiratory, and cerebrovascular diseases in several geographic areas [6,48,62]. MP weather has been associated with increased mortality by influenza and pneumonia around 15–19 days afterwards [63]. Likewise, as the cold weather types, particularly DP, are positively associated with the occurrence of influenza, that could increase total mortality rates; on the other hand, influenza incidences are reportedly lower in tropical weathers (DT and MT) [64]. 

### 4.3. Strengths and Limitations

Most previous studies just analyzed the effect of temperature on health or mortality and mainly in mid-latitude locations. This is the first study assessing the relationship between oppressive weather types based on SSC and all-cause mortality in northern and southern locations in Sweden but also in two seasons, covering a 24-year period from 1991 to 2014.

The large sample size (number of deaths) in southern locations allowed us to obtain more precise results. However, the study has some limitations that must be addressed. The small number of daily deaths in northern locations (mean counts in Jämtland, four and in Västerbotten, five) produced some large effects, but these were generally not statistically significant with wide confidence intervals and thus low precision of estimates. On the other hand, in winter, the number of days with DT and MT weather types were small, more limiting especially in the northern locations. 

## 5. Conclusions

In the present study, the association between total-cause mortality and oppressive weather was assessed during summer and winter in southern and northern locations in Sweden. Differences in mortality associated to oppressive weather were found between southern and northern locations and across the seasons.

Dry Tropical (DT) and Moist Tropical (MT) were the most harmful oppressive weather types, affecting mainly the southern locations during the summer. Nevertheless, even Dry Polar (DP) as well as Moist Polar (MP) weather types in summer increased the mortality at longer lags in the southern study areas. The northern study locations were less affected by oppressive weather types in both summer and winter.

The SSC is a useful tool to identify short-term effects of weather types on mortality. The association between oppressive weather types and mortality indicate that there are other factors rather than temperature that affect mortality. However, the small population in northern locations limits the precision of the estimate.

This study could contribute to improving weather-health warning systems considering the use of SSC instead of temperature, as these results could contribute to improving public health preparedness and response measures.

The impact of different SSC weather types on specific causes of death (e.g., cardiovascular and respiratory mortality) as well as hospitalizations in Sweden will be studied in the near future.

## Figures and Tables

**Figure 1 ijerph-16-01696-f001:**
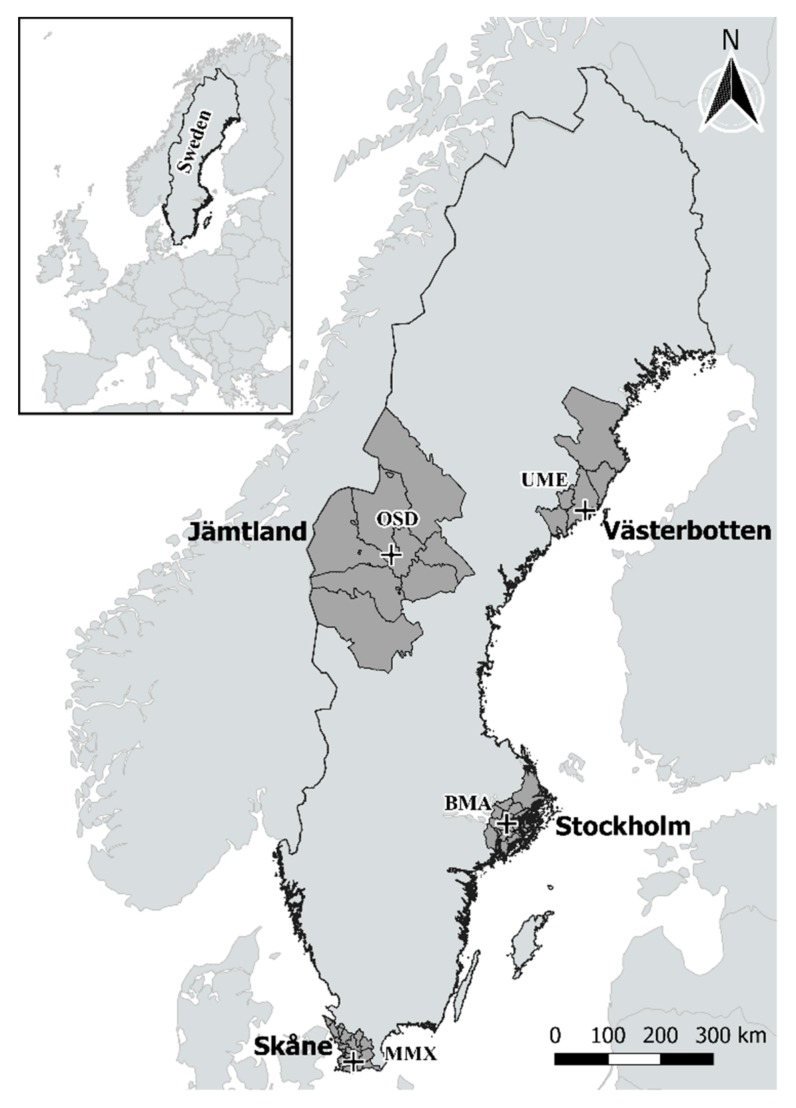
Map of Sweden with study locations. The four study regions are in dark gray. Locations of main weather stations in each study area are represented by black crosses (Malmö—MMX, Bromma—BMA, Östersund—OSD, and Umeå—UME).

**Figure 2 ijerph-16-01696-f002:**
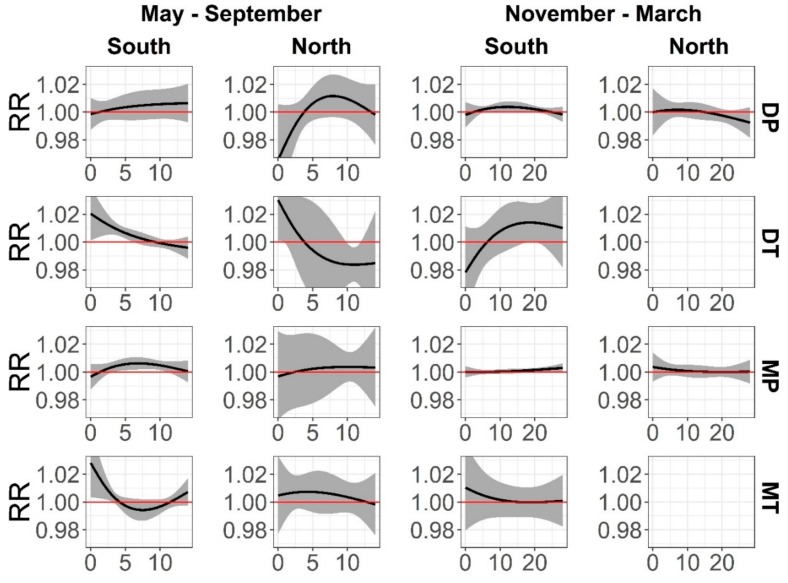
RR (with 95% CI) of all-cause mortality by oppressive weather type at different lags from 1991 to 2014 for summer (May–September) and winter (November–March), respectively. RR and 95% CI for DT and MT weather types could not be estimated for northern locations because of their rare occurrence during the winter.

**Table 1 ijerph-16-01696-t001:** Characteristics of each weather type for January and July at each location.

Weather Type	Months	Skåne (South West) MMX				Stockholm (South East) BMA				Jämtland (North West) OSD				Västerbotten (North East) UME			
		%	Ta	Tp	Td	%	Ta	Tp	Td	%	Ta	Tp	Td	%	Ta	Tp	Td
**DP**	JAN	6	−6	−5	−9	15	−9	−8	−11	12	−18	−18	−21	24	−15	−15	−17
	JUL	2	11	17	7	3	12	18	7	5	7	14	4	2	8	14	6
**DM**	JAN	10	2	2	−2	11	1	1	−3	13	−1	−1	−6	10	−1	0	−4
	JUL	21	11	21	10	27	12	22	9	15	10	19	7	27	10	19	8
**DT**	JAN	0	-	-	-	0	-	-	-	0	-	-	-	0	-	-	-
	JUL	12	13	25	12	14	15	27	10	9	14	25	9	8	13	24	11
**MP**	JAN	25	−4	−3	−4	30	−4	−4	−5	31	−11	−10	−12	31	−9	−9	−10
	JUL	16	12	15	11	6	11	14	10	11	8	11	6	2	9	11	9
**MM**	JAN	45	2	3	2	32	1	2	0	33	−2	−1	−3	24	−1	−1	−2
	JUL	33	14	19	13	31	14	18	13	43	11	16	10	43	13	17	12
**MT**	JAN	5	6	7	5	<1	6	7	4	0	-	-	-	0	-	-	-
	JUL	9	17	24	15	13	17	24	14	9	15	22	11	12	16	22	14
**TR**	JAN	8	−4	−1	−3	12	−4	−2	−4	11	−7	−5	−8	11	−9	−5	−8
	JUL	7	14	19	11	7	14	21	10	8	11	16	8	7	12	18	9

DP = Dry Polar, DM = Dry Moderate, DT = Dry Tropical, MP = Moist Polar, MM = Moist Moderate, MT = Moist Tropical, TR = Transition. % = Percentage of days in month classified as this weather type, Ta = temperature at 3h (°C), Tp = temperature at 15h (°C), Td = Dew point at 15h (°C). Weather station named in the columns of each study location (Malmö—MMX, Bromma airport—BMA, Östersund—OSD and Umeå airport—UME). Detailed information about the characteristics of each weather type regarding the month and location are described at: http://sheridan.geog.kent.edu/ssc.html.

**Table 2 ijerph-16-01696-t002:** Descriptive statistics of weather types and death counts in the four study locations, 1991–2014.

Location		Weather Type						
		DM	DP	DT	MM	MP	MT	TR
**Skåne** **(South west)**	**Number of days** (%)	1653 (18.9)	625 (7.1)	359 (4.1)	3408 (38.9)	1618 (18.5)	479 (5.5)	624 (7.1)
	**Total number of deaths** (%)	37209 (18.3)	14860 (7.3)	8065 (4.0)	79282 (39.0)	38184 (18.8)	11193 (5.5)	14476 (7.1)
	**Mean daily number of** **deaths (±SD)**	22.5 (5.1)	23.8 (5.3)	22.5 (5.1)	23.3 (5.2)	23.6 (5.5)	23.4 (4.7)	23.2 (5.1)
**Stockholm** **(South East)**	**Number of days** (%)	1966 (22.4)	1154 (13.2)	433 (4.9)	2254 (25.7)	1756 (20.1)	430 (4.9)	765 (8.7)
	**Total number of deaths** (%)	80301 (21.9)	48984 (13.3)	17460 (4.8)	94707 (25.8)	75191 (20.5)	17573 (4.8)	32730 (8.9)
	**Mean daily number of** **deaths (±SD)**	40.8 (7.3)	42.4 (7.5)	40.3 (6.9)	42 (7.3)	42.8 (7.4)	40.9 (7.2)	42.8 (13.2)
**Jämtland** **(North West)**	**Number of days** (%)	1556 (17.8)	819 (9.3)	247 (2.8)	3184 (36.3)	1950 (22.3)	263 (3.0)	741 (8.5)
	**Total number of deaths** (%)	6885 (17.8)	3646 (9.4)	1033 (2.7)	13792 (35.7)	8853 (22.9)	1053 (2.7)	3398 (8.8)
	**Mean daily number of** **deaths (±SD)**	4.4 (2.2)	4.5 (2.2)	4.2 (2.1)	4.3 (2.1)	4.5 (2.2)	4.0 (2.0)	4.6 (2.2)
**Västerbotten** **(North East)**	**Number of days** (%)	1795 (20.5)	1459 (16.7)	194 (2.2)	2644 (30.3)	1496 (17.1)	340 (3.9)	812 (9.3)
	**Total number of deaths**(%)	8751 (19.9)	7567 (17.2)	958 (2.2)	13090 (29.7)	7902 (17.9)	1619 (3.7)	4180 (9.5)
	**Mean daily number of** **deaths (±SD)**	4.9 (2.2)	5.2 (2.4)	4.9 (2.4)	5.0 (2.3)	5.3 (2.4)	4.8 (2.2)	5.1 (2.4)

DP = Dry Polar, DM = Dry Moderate, DT = Dry Tropical, MP = Moist Polar, MM = Moist Moderate, MT = Moist Tropical, TR = Transition.

**Table 3 ijerph-16-01696-t003:** Cumulative Relative Risk and 95% confidence intervals for all-cause mortality over 14-days for summer and 28-days for winter.

Oppressive Weather	May–September		November–March	
	South	North	South	North
DP	1.06 (0.94–1.20)	1.00 (0.90–1.11)	**1.05 (1.01–1.09)**	0.98 (0.88–1.08)
DT	**1.08 (1.02–1.14)**	0.93 (0.76–1.14)	1.40 (0.87–2.23)	-
MP	**1.05 (1.01–1.09)**	1.03 (0.86–1.25)	1.03 (1.00–1.06)	1.02 (0.95–1.11)
MT	**1.05 (1.01–1.1** **0)**	1.06 (0.93–1.21)	1.06 (0.71–1.60)	-

Values in bold represent statistically significant relative risk. DP = Dry Polar, DT = Dry Tropical, MP = Moist Polar, MT = Moist Tropical.

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
