# Peer review of "Association between Weather Types based on the Spatial Synoptic Classification and All-Cause Mortality in Sweden, 1991–2014"

_ijerph, 2019, doi:10.3390/ijerph16101696_

Round 1

Reviewer 1 Report

The manuscript studies the relationship between synoptic airmass categories and all-cause mortality in several locations of Sweden. The impact is estimated using a DLNM to account for the delay between an exposure to one of the oppressive weather type and the mortality response.

It is an overall well-conducted study with the analysis addressing the claimed objectives. The paper is also properly structured and well written. In addition to the study being the first considering synoptic airmass categories in Sweden, the use of SSC with a DLNM also seems to be original. Therefore, I definitely think that the paper has its place in IJERPH.

I nonetheless have a few reservations concerning the use of the SSC in such a context, but it could come from my limited knowledge about SSC. In the following, the “major” comments are aspects that should be addressed (or at least discussed) by the author in their revision, while “minor” comments rather concern details.

Major comments

-          How is the estimation of the DLNM conducted when considering airmass categories as the main predictor? If I am not mistaken, it is a categorical variable, which is unusual in this context. Is there one coefficient per category ? And I guess that RRs are computed by dividing the estimated relationship of the oppressive weather categories by those of the other categories, but it is not clearly stated. A bit of mathematical details could be of help here.

-          For the sake of the interpretation and the discussion, it would be useful to have more details about the results of the SSC. For instance the range of Temperature, humidity and Wind Speed of each category, but also a bit of interpretation. Some results would perhaps make more sense to readers unfamiliar with SSC. For instance, it is not clear to me what could a “Polar” climate represent during summer.

-          It would be interesting to compare the result reported in the manuscript to the current knowledge about weather-related mortality. It seems to me that the RRs reported in the manuscript are quite low (e.g. compared to the curve for Stockholm in Gasparrini et al., 2015).

-          It leads to a comment that is more an overture to discussion: are airmass categories good predictors of mortality in the location considered? For all I know (not much actually), health does not intervene in the construction of the categories.  

Minot comments

-          Section 2.1 : There is a typo in the title “Statistical analysis”

-          Why were the months of October and April discarded from the analysis?

-          Lines 148-152: Note that instead of manually testing different parametrizations, the penalized version of the DLNM could be used (Gasparrini et al., 2017).

-          Lines 154 and 156: I would personally prefer mathematical expressions for the models, it would be clearer (e.g. what is weather exactly?).

-          Figure 2: Why are there no functions for MT or DT in northern locations?

-          I suppose the health issue is all cause mortality, but it is never stated.

Author Response

Reviewer 1

Comments and Suggestions for Authors

The manuscript studies the relationship between synoptic airmass categories and all-cause mortality in several locations of Sweden. The impact is estimated using a DLNM to account for the delay between an exposure to one of the oppressive weather type and the mortality response.

It is an overall well-conducted study with the analysis addressing the claimed objectives. The paper is also properly structured and well written. In addition to the study being the first considering synoptic airmass categories in Sweden, the use of SSC with a DLNM also seems to be original. Therefore, I definitely think that the paper has its place in IJERPH.

I nonetheless have a few reservations concerning the use of the SSC in such a context, but it could come from my limited knowledge about SSC. In the following, the “major” comments are aspects that should be addressed (or at least discussed) by the author in their revision, while “minor” comments rather concern details.

Major comments

-          How is the estimation of the DLNM conducted when considering airmass categories as the main predictor? If I am not mistaken, it is a categorical variable, which is unusual in this context. Is there one coefficient per category ? And I guess that RRs are computed by dividing the estimated relationship of the oppressive weather categories by those of the other categories, but it is not clearly stated. A bit of mathematical details could be of help here.

->  We agree, the formulae was included replacing the R code and the explanation was improved (See lines 157 – 164 of the revised manuscript)

-          For the sake of the interpretation and the discussion, it would be useful to have more details about the results of the SSC. For instance the range of Temperature, humidity and Wind Speed of each category, but also a bit of interpretation. Some results would perhaps make more sense to readers unfamiliar with SSC. For instance, it is not clear to me what could a “Polar” climate represent during summer.

->  Table 1 shows the summarized characteristics of each weather type in January (Winter) and July (Summer) in the study locations. In the column of each location, the acronyms of the weather stations were included; also the address (http://sheridan.geog.kent.edu/ssc.html) where detailed explanation of the classification procedure description of each weather type was added in the foot notes of Table 1 (See lines 131 – 134 of the revised manuscript).

In the results section is provided the information about temperature (Temp) and dew point (Td) of each weather type (See lines 187 – 205 of the revised manuscript)

-          It would be interesting to compare the result reported in the manuscript to the current knowledge about weather-related mortality. It seems to me that the RRs reported in the manuscript are quite low (e.g. compared to the curve for Stockholm in Gasparrini et al., 2015).

->  We agree, the differences between our results and the results obtained by Gasparrini et al., 2015 were discussed in the revised manuscript (See lines 252 – 257 of the revised manuscript).

-          It leads to a comment that is more an overture to discussion: are airmass categories good predictors of mortality in the location considered? For all I know (not much actually), health does not intervene in the construction of the categories. 

->  As you mentioned, health does not intervene in the construction of the categories. The construction of the SSC categories is based on weather variables (temperature, dew-point temperature, pressure, wind speed, and cloud opacity). The methodology was described by Sheridan (2002)  [1] (See lines 67 – 73 of the revised manuscript). According to Hondula et al, 2014 [2] has been used widely in health-related studies. Also, several heat-health warning systems around the world are based on SSC [3, 4]. In our study we demonstrated the association between different weather types and mortality. However, the small sample sizes in northern locations could affect the precision of the results in this area (See lines 230 – 231 of the revised manuscript)

Minor comments

-          Section 2.1 : There is a typo in the title “Statistical analysis”

-> We agree, the mistake was corrected (See line 135 of the revised manuscript)

-          Why were the months of October and April discarded from the analysis?

-> October and April were discarded form the analysis because during these months the weather variability is high, and extremes in either direction are rarer.  We thus focused on the core areas of summer and winter, extending these outwards to capture heat and cold events slightly outside the traditional boundaries; since we found almost none in October and April, these were excluded.  

-          Lines 148-152: Note that instead of manually testing different parametrizations, the penalized version of the DLNM could be used (Gasparrini et al., 2017).

-> Thank you very much for this information and surely it will be used in the next studies making easier this process.

-          Lines 154 and 156: I would personally prefer mathematical expressions for the models, it would be clearer (e.g. what is weather exactly?).

->  We agree, the R code was removed and the mathematical expression was presented (See lines 157 – 172 of the revised manuscript)

-          Figure 2: Why are there no functions for MT or DT in northern locations?

->  In Sweden, the study northern locations during the winter in some years don’t have days classified as Moist Tropical or Dry Tropical or at least there are very few days of MT or DT. Therefore,  it is not possible to estimate the RR. (See lines 199 and 200 in the revised manuscript). The Figure 2 was modified removing the horizontal red lines form the plots of MT and DT in northern locations during the winter. Also, the regarding explanation was added in the title of the figure as follow: “RR and 95%CI for DT and MT weather types could not be estimated for northern locations because of their rare occurrence during the winter”.

-          I suppose the health issue is all cause mortality, but it is never stated.

-> Yes, the health issue is the all-cause mortality (see lines 85, 160, 189, 215, among others in the revised manuscript.)

Dear reviewer, thank you for your valuable comments and effort to improve the manuscript. We considered carefully all your recommendation.

References

1.           Sheridan, S. C., The redevelopment of a weather-type classification scheme for North America. International Journal of Climatology 2002, 22, (1), 51-68.

2.           Hondula, D. M.; Vanos, J. K.; Gosling, S. N., The SSC: a decade of climate-health research and future directions. Int J Biometeorol 2014, 58, (2), 109-20.

3.           Kalkstein, A. J.; Sheridan, S. C., The social impacts of the heat-health watch/warning system in Phoenix, Arizona: assessing the perceived risk and response of the public. International Journal of Biometeorology 2007, 52, (1), 43-55.

4.           Sheridan, S. C.; Kalkstein, L. S., Progress in heat watch-warning system technology. Bulletin of the American Meteorological Society 2004, 85, (12), 1931-+.

Reviewer 2 Report

IJERPH-493262: Association between weather types based on the Spatial Synoptic Classification and all-cause mortality in Sweden, 1991 – 2014

The topic of the paper is worthwhile with good approach. The structure, presentation and synthesis of findings conform to many of the scientific standards along with the relevance of the topic with current challenges related to climate change. There are few observations which need further attention.

The first paragraph of introduction seems wayward at some places and I would recommend to offer one more read by the authors for possible language issues.

The methods are well discussed and suited to the data and results are interesting. The discussion of results is well in place but the conclusions need to be concise, informative and offer some valid implications for the region and research.

Minor: Line 115: ‘were’ instead of was. Line 139: typo or punctuation.

Author Response

Reviewer 2

Comments and Suggestions for Authors

IJERPH-493262: Association between weather types based on the Spatial Synoptic Classification and all-cause mortality in Sweden, 1991 – 2014

The topic of the paper is worthwhile with good approach. The structure, presentation and synthesis of findings conform to many of the scientific standards along with the relevance of the topic with current challenges related to climate change. There are few observations which need further attention.

The first paragraph of introduction seems wayward at some places and I would recommend to offer one more read by the authors for possible language issues.

->  We agree, the first paragraph was revised and corrected

The methods are well discussed and suited to the data and results are interesting. The discussion of results is well in place but the conclusions need to be concise, informative and offer some valid implications for the region and research.

 ->  We agree, the conclusions were modified according to your suggestion

Minor: Line 115: ‘were’ instead of was. Line 139: typo or punctuation.

->  We agree. In line 115 (See line 117 of the revised manuscript), “was” was replaced by “were”. We modify the text in line 139 (See line 143 of the revised manuscript) to solve the mistake.

Dear reviewer, we appreciate all your insightful comments. Thank you for taking the time and effort to help us improve the manuscript.
